# Empirical mean-noise fitness landscapes reveal the fitness impact of gene expression noise

Jörn M. Schmiedel [1], Lucas B. Carey [2,3] & Ben Lehner [1,4,5]

The effects of cell-to-cell variation (noise) in gene expression have proven difficult to quantify because of the mechanistic coupling of noise to mean expression. To independently quantify the effects of changes in mean expression and noise we determine the fitness landscapes in mean-noise expression space for 33 genes in yeast. For most genes, short-lived (noise) deviations away from the expression optimum are nearly as detrimental as sustained (mean) deviations. Fitness landscapes can be classified by a combination of each gene's sensitivity to protein shortage or surplus. We use this classification to explore evolutionary scenarios for gene expression and find that certain landscape topologies can break the mechanistic coupling of mean and noise, thus promoting independent optimization of both properties. These results demonstrate that noise is detrimental for many genes and reveal non-trivial consequences of mean-noise-fitness topologies for the evolution of gene expression systems.

[1] Centre for Genomic Regulation (CRG), The Barcelona Institute of Science and Technology, Doctor Aiguader 88, 08003 Barcelona, Spain. [2] Department of Experimental and Health Sciences, Universitat Pompeu Fabra, Doctor Aiguader 88, 08003 Barcelona, Spain. [3] Center for Quantitative Biology and Peking-Tsinghua Center for the Life Sciences, Academy for Advanced Interdisciplinary Studies, Peking University, Beijing 100871, China. [4] Universitat Pompeu Fabra (UPF), 08003 Barcelona, Spain. [5] ICREA, Passeig Lluís Companys 23, 08010 Barcelona, Spain. Correspondence and requests for materials should be addressed to J.M.S. (email: joern.schmiedel@gmail.com) or to B.L. (email: ben.lehner@crg.eu)

The mapping between genotype and phenotype determines how genetic variation affects phenotypes and how in turn genotypes evolve under natural selection. An important molecular phenotype for each gene is its protein abundance (Fig. 1a). Protein abundances are tightly controlled at multiple regulatory levels. They do, however, show considerable variation, not only across genotypes and environments, but also among isogenic cells within the same environment and in the same cell over time[1,2]. This non-genetic variation in protein abundances results from the stochasticity of production and degradation reactions as well as from the variable abundances of regulators[3–5], with time-scales of such fluctuations often on the order of one or two cell cycles[6].

A gene's protein abundance distribution is commonly characterized by its average (mean) and width (noise). Mean and noise of protein abundance distributions are, however, not independent quantities, but are instead mechanistically coupled by the protein production process. In particular, switching between transcriptional permissive and prohibitive states leads to proteins being produced in bursts. While the size of bursts (the rates at which mRNAs and proteins are produced in the permissive state and how quickly genes revert back to a transcriptionally prohibitive state) only affects mean protein abundances, the frequency of bursts affects mean protein abundances and noise in an inversely proportional manner[7–11]. Mutations in promoters most often affect burst frequencies, resulting in negatively correlated changes in mean and noise[7]. A negative correlation between mean abundances and noise is also observed across genes[12–14].

Both large[15–21] as well as small[22–25] sustained deviations of mean protein abundance from levels that maximize fitness have been found to be detrimental to organismal fitness.

The fitness effects of noise in protein abundances are less well explored. One can distinguish two scenarios. If mean protein abundance is far from the level that maximizes fitness, high noise can be beneficial by allowing some cells to transiently express more optimal protein abundances[26]. In fluctuating environments, high expression noise may therefore be a bet-hedging strategy to diversify phenotypes[27–31].

If mean protein abundance is, however, close to the level that maximizes fitness, as is presumably the case for many genes in more stable environments[24], then high noise should be detrimental because fluctuations result in sub-optimal protein abundance. The observed low noise levels of many dosage-sensitive genes in yeast provide circumstantial evidence that too much noise is detrimental and has been selected against[7,12,13,24,32–35]. However, the mechanistic coupling of noise and mean levels in protein production has made it difficult to directly test the fitness consequences of changes in expression noise alone. Notably, the Wittkopp lab has recently demonstrated that yeast strains in which the TDH3 gene, for which deviations in mean abundances away from wild-type levels are detrimental to yeast fitness[36], is driven by high noise promoters are less fit[26]. Consequently, noise-increasing mutations in its endogenous promoter have been found to be under purifying selection[37].

Whether these results for TDH3 generalize to other genes is, however, unclear. Importantly, we still lack quantitative experimental data and understanding of the fitness effects of expression noise and its relationship to the optimality of mean protein abundances (Fig. 1b). Therefore, how these two expression phenotypes might co-evolve, especially given their mechanistic couplings by the transcriptional process, is still an open question (Fig. 1c).

Here we reconstruct fitness landscapes in mean-noise expression space for 33 genes in yeast using published fitness data of yeast strains in which genes are driven by a library of synthetic promoters[24,38] (Fig. 2a–c). These continuous landscapes allow for a comprehensive, quantitative assessment of both the independent as well as interdependent fitness effects of noise and mean expression. Overall, half of the assayed genes are noise intolerant and the fitness impact of increased noise is nearly as detrimental as equivalent changes in mean expression away from optimum. Principal component analysis of mean-noise-fitness landscapes reveals that the landscapes can be decomposed into two principal landscape topologies, representing sensitivity of fitness towards protein shortage or surplus. These two principal topologies link the fitness effects of mean deviations and noise and thus determine how intolerant a gene is to high expression noise. We further use the expression-fitness landscapes to explore how mean and noise can evolve, given their mechanistic coupling imposed by the transcriptional process. We find that on landscapes of genes sensitive to both protein shortage and surplus the mechanistic coupling between mean and noise is broken, therefore allowing for the independent minimization of noise levels.

Together, our analyses reveal the quantitative fitness effects of expression noise and their relation to mean expression and how the evolution of gene expression is shaped by the interplay between phenotypic constraints and expression-fitness landscape topology.

## Results

**Reconstruction of fitness landscapes in mean-noise space.** We obtained data on the fitness of yeast strains where in each strain one of a panel of 85 genes is driven by one of a panel of 120 synthetic promoters[24]. Here, in one set of experiments, the library of 120 synthetic promoters was cloned upstream of each of 85 open reading frames, replacing the endogenous promoter (Fig. 2a). All constructed strains were pooled and their fitness (growth rate in glucose) was measured in competitive growth experiments. In a second set of experiments, the synthetic promoters as well as the endogenous promoters of all investigated genes were cloned in front of YFP in the HIS3 locus and flow cytometry was used to determine their relative mean expression strength (Fig. 2b). Together, this allowed the authors to analyze the fitness effects of mean expression changes relative to the wild-type expression of genes[24].

In addition to this dataset, we also obtained data from an earlier study[38] from the same group of authors that measured both mean and cell-to-cell variation (noise, coefficient of variation, i.e., the standard deviation divided by the mean) in the expression of the same set of synthetic promoters driving YFP on a plasmid (Fig. 2c). This was achieved by sorting cells along the overall expression distribution, reconstructing individual promoter expression distributions from deep sequencing of sorted cell populations and quantifying their mean and noise.

When combined, these data allow us to not only assess how the mean but also the shape (as quantified here by mean and noise) of protein abundance distributions affects fitness by comparing strains in which different promoters drive the same gene. While the absolute expression strength and noise of a particular promoter can depend on its genomic location, for the following analyses we make the assumption that the *relative* expression strength and noise levels between promoters is independent of the genomic location. The validity of this assumption is supported by the literature[39,40] as well as by the high correlation of mean expression strengths when the synthetic promoters are driving YFP from a plasmid or the HIS3 locus ($R^2 = 0.93$, $R^2 = 0.99$ after pre-processing, i.e., exclusion of 11 outliers, see Supplementary Fig. 1a and "Methods").

We filtered the set of promoters used in the original studies according to several quality control criteria and in order to obtain

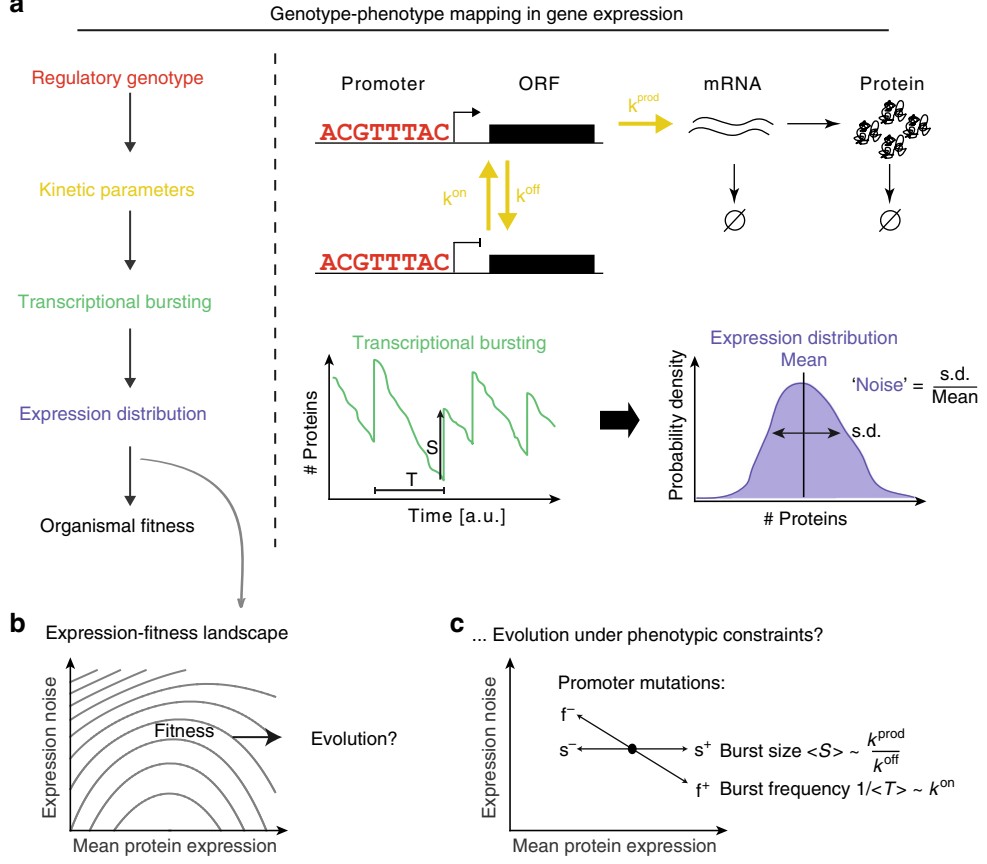

**Fig. 1** Genotype-phenotype mapping in expression and its link to organismal fitness. **a** Multi-level mapping from regulatory genotype to expression phenotypes to organismal fitness. Regulatory genotypes (red) determine kinetic parameters (e.g. promoter on/off switching $k^{on}/k^{off}$, rate of mRNA transcription $k^{prod}$, yellow), thus the properties of transcriptional bursting (average size $<S>$ and frequency $1/<T>$, green), which results in particular protein expression distributions (blue). Expression distributions, characterized by their mean and noise (normalized standard deviation, i.e. coefficient of variation), determine organismal fitness via the expression-fitness landscape (panel **b**). **b** The expression-fitness landscape (contours depict equi-fitness levels) describes the mapping between the expression distribution of a gene (i.e. its properties mean and noise) and fitness. Quantifying the topology of expression-fitness landscapes may inform about the evolution of gene expression systems. **c** The transcriptional process constrains how genetic variation can move gene expression in mean-noise space; e.g. promoter mutations cannot lead to purely vertical moves (i.e. only changing noise). Does this affect the evolution of gene expression systems?

a homogenously populated region in the expression mean-noise space (Supplementary Fig. 1, and "Methods"). In the final dataset, each gene is expressed under the control of 74 to 79 (average 78) different synthetic promoters that span an expression range of 16-fold and a noise range of 4-fold, as assessed by the coefficient of variation (Fig. 2d).

To systematically study the fitness effects of varying both mean and noise around wild-type expression levels we restricted our analyses to the 33 genes with wild-type expression levels in the centre of the well-populated mean-noise space (Supplementary Fig. 1d, see "Discussion" for consideration of effects away from wild-type levels). These genes represent a wide range of cellular functions, such as transcription factors, RNA polymerase, proteasome, cytoskeleton, trafficking and metabolism.

We started our analysis by examining the fitness of gene-promoter strains in the expression mean-noise space (Fig. 2d and Supplementary Fig. 2). For some genes, like the topoisomerase I *TOP1*, all strains across the mean-noise space have approximately wild-type fitness. In contrast, for the 26S proteasome subunit *RPN8* strains with low expression (and high noise) promoters tend to have low fitness. Additionally, for the beta-tubulin *TUB2*, strains with high expression and high noise promoters also have lower fitness.

We sought a systematic way to investigate how mean and noise impact fitness, both together and independently. We reasoned that for each gene there exists a continuous fitness landscape in the mean-noise expression space. This landscape has been experimentally sampled by the different synthetic promoter strains.

To reconstruct a smooth, continuous fitness landscape for each gene we calculated fitness values on a regular grid across the mean-noise space using a Gaussian smoothing approach. For each point on the grid a fitness value was calculated as the weighted sum of all measured fitness values for that gene. Weights were calculated according to a bivariate normal kernel (Fig. 2e), centred on the grid-point and with gene-independent scaling parameters in mean and noise direction optimized to minimize the root mean squared error between the smoothed fitness landscapes and the raw data (estimated using ten-fold cross-validation). The weighting of each synthetic promoter strain was further modified by the measurement error of its mean expression, noise and fitness values (see "Methods" for full details). This weighted smoothing across fitness measurements from independent promoter strains results in low uncertainty of fitness values across the landscape, up to four times lower than the overall variability of fitness values across landscapes (Supplementary Fig. 3).

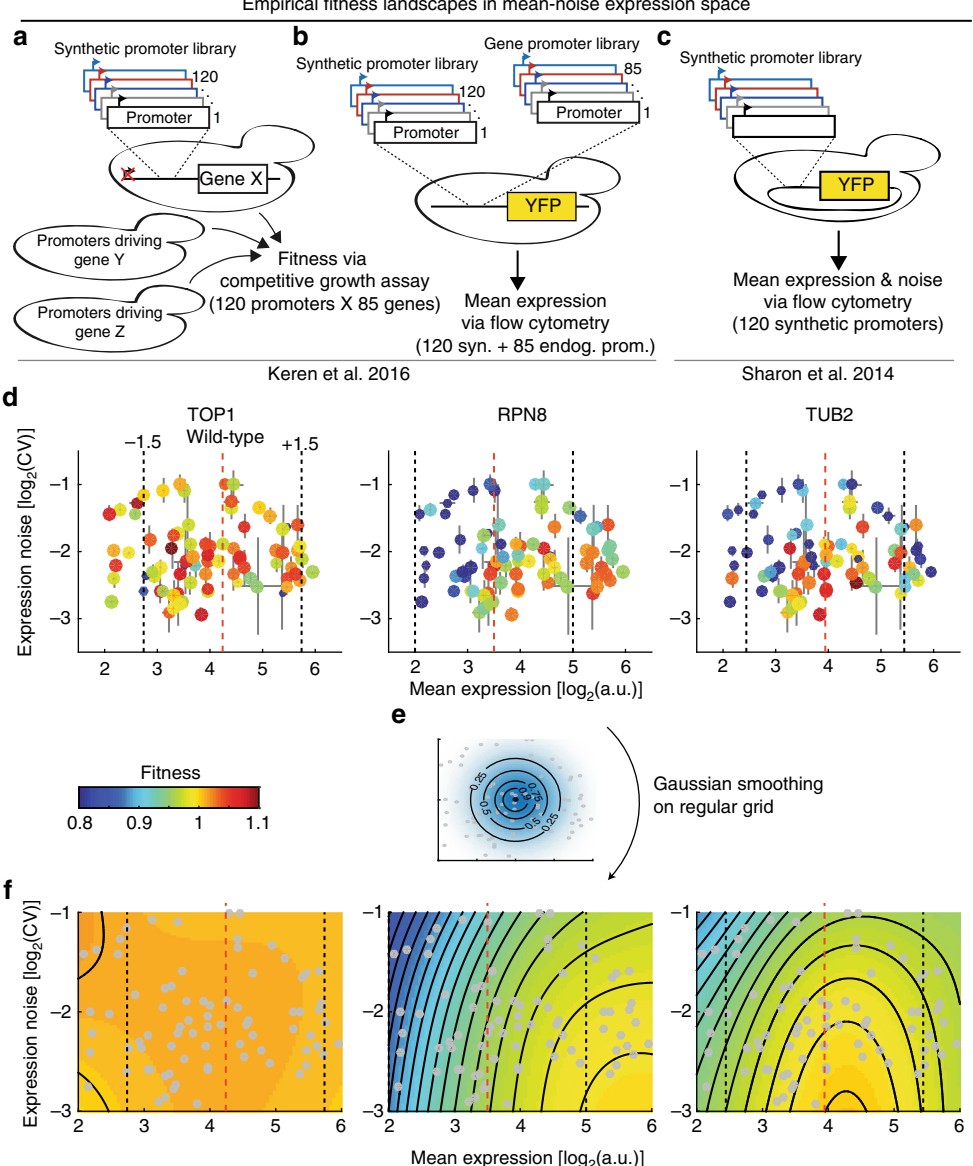

**Fig. 2** Mean-noise-fitness landscapes reveal gene-specific responses to expression deviations. **a** Keren, et al.[24] constructed yeast strains where in each strain one of a panel of genes (X, Y, Z,...) is driven by one of a panel of 120 synthetic promoters. All constructed strains were pooled and their fitness was measured in competitive growth experiments. **b** Relative mean expression strength of the synthetic promoters as well as the endogenous promoters of all investigated genes was measured by cloning each promoter in front of *YFP* in the *HIS3* locus and assaying the resulting strains individually by flow cytometry[24]. **c** Sharon, et al.[38] cloned synthetic promoters in front of *YFP* on a plasmid. Fluorescent activated cell sorting in combination with deep sequencing was used to infer mean expression and noise of individual promoters. **d** Fitness of synthetic promoter-gene strains plotted in mean—noise expression space for three example genes. Each panel shows the fitness (growth rate relative to wild-type, indicated by colour) of 79 yeast strains, in each of which a particular synthetic promoter drives the indicated gene, as a function of mean expression and noise of the synthetic promoters. Size of circles is inversely proportional to the measurement error of fitness. Error bars in horizontal and vertical directions indicate measurement error (s.e.m.) in mean and noise direction, respectively. Red vertical line shows estimated wild-type expression of the gene, dashed vertical lines mark region ± 1.5-fold from wild-type expression. Source data are provided as a Source Data file. **e**, Mean-noise-fitness landscapes were reconstructed using Gaussian smoothing on a regular grid, with fitness at each grid point as the weighted sum over the fitness of all strains. Weighting depends on distance to grid point (shown) and measurement errors in all three dimensions (see Methods). **f** Smoothed mean-noise-fitness landscapes of the three genes shown in panel **d**. Surface colour indicates fitness. Contour lines are spaced in increments of 0.01 fitness units. Strains (grey points), wild-type expression (red vertical line) and ± 1.5-fold around wild-type expression (grey vertical lines) are indicated. CV coefficient of variation, a.u. arbitrary units

The reconstructed mean-noise fitness landscapes reveal that for *TOP1* there is essentially no systematic effect of mean expression or expression noise on fitness (Fig. 2f, see Fig. 3 for all landscapes). The fitness landscape of *RPN8* reveals coupled negative fitness effects of lowered mean expression and high noise. Finally, the fitness landscape of *TUB2* reveals a non-linear relationship between mean and noise on fitness. High expression noise is always detrimental, but the effect of noise on fitness increases as mean expression deviates more from the wild-type expression level.

Together, expression-fitness landscapes in the mean-noise space thus present a valuable opportunity to study the interplay

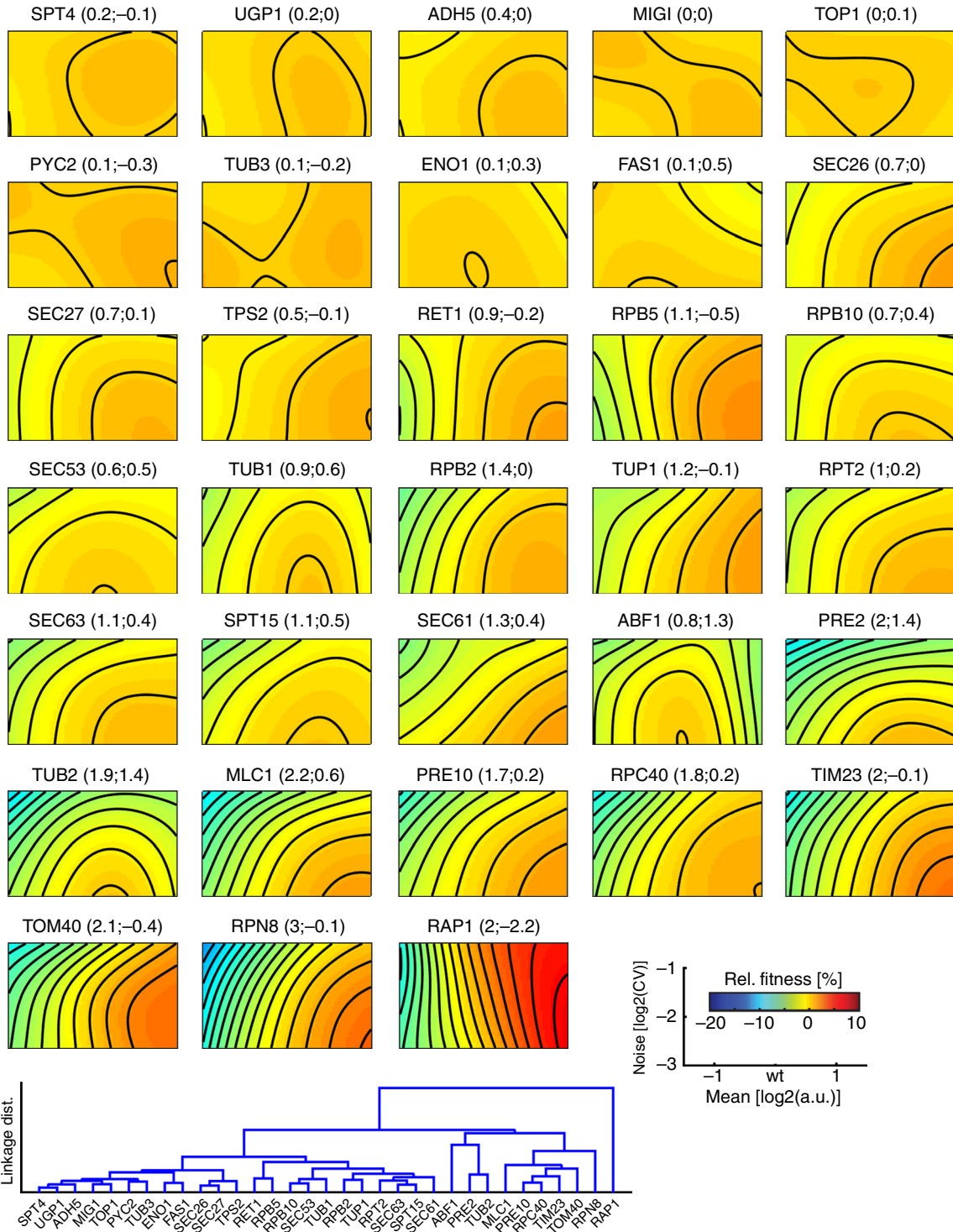

**Fig. 3** Expression-fitness landscapes in mean-noise space around wild-type mean expression. Expression-fitness landscapes for the investigated 33 yeast genes. Landscapes are ordered according to hierarchical clustering of Euclidean distances between fitness profiles (see linkage graph in lower left). Mean expression is limited to ±1.5 log₂-units around genes' wild-type mean expression. Colours indicate fitness relative to wild-type mean expression and lowest noise level; contours are spaced in increments of 0.01 fitness units. Plot titles give gene name and principal topology 1 and 2 loadings, respectively (related to Fig. 5a)

of two molecular phenotypes of gene expression on fitness, in a quantitative and systematic manner.

**High noise is as detrimental as non-optimal mean expression.** We first quantified the effects of changes in mean and noise on fitness and their relationship across individual fitness landscapes. We calculated for each gene the effect of mean expression changes on fitness, its *expression sensitivity*, as the average fitness loss upon a two-fold change in mean expression at minimal expression noise levels (Fig. 4a). Equivalently, we quantified for each gene the fitness effect of expression noise, its *noise intolerance*, as the average fitness loss upon a twofold increase in noise at wild-type mean expression (Fig. 4a). Importantly, assessment of both quantities is robust to the exact metric chosen (Supplementary Fig. 4a).

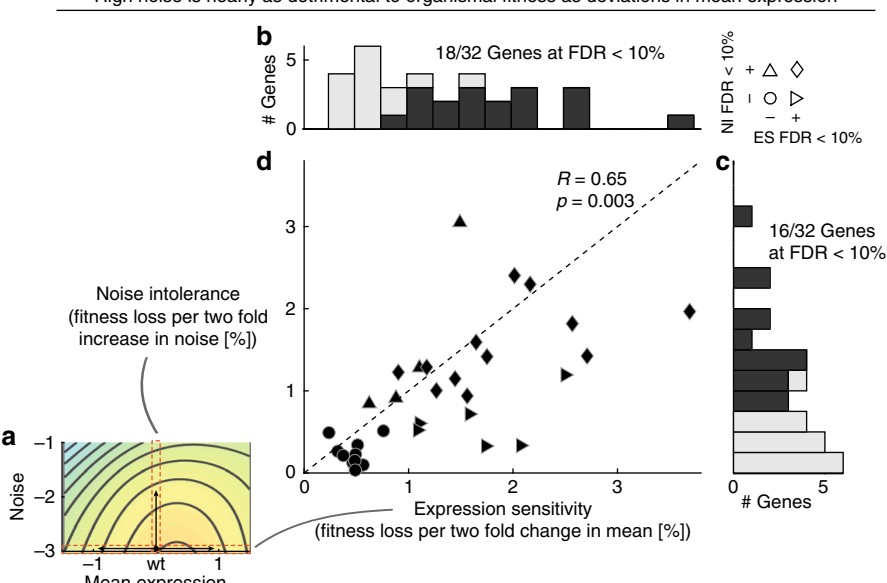

**Fig. 4** High noise is nearly as detrimental as non-optimal mean expression. **a** A gene's expression sensitivity is the loss of fitness upon two-fold changes in mean expression at low expression noise (indicated by horizontal red bar). Its noise intolerance is the loss of fitness upon a two-fold increase noise levels at wild-type mean expression (indicated by vertical red bar). **b** Distribution of expression-sensitivity across 32 genes, of which 18 are significantly sensitive at false discovery rate < 10% (dark colour). **c** Distribution of noise intolerant across 32 genes of which 16 are significantly intolerant at false discovery rate < 10% (dark colour). **d** Relationship between expression sensitivity and noise intolerance across all genes. Pearson correlation coefficient and $p$, the fraction of $10^4$ sets of randomized landscapes that have Pearson correlation coefficients greater than the one found on the real landscapes, are indicated. Shapes indicate false discovery rate combinations of genes with respect to expression sensitivity and noise intolerance, as shown in legend in upper right corner of the plot. Note that results for *RAP1* are not shown and are discussed in the Supplementary Note 1. Source data are provided as a Source Data file

A two-fold change in mean expression levels results in fitness losses from 0.3% to 3.7%, with an average of 1.4% across landscapes (Fig. 4b). More than half of the assayed genes (19 out of 33) are significantly expression sensitive (at false discovery rate (FDR) < 10%, estimated using randomized control landscapes) and the estimated expression sensitivities of genes are highly predictive of known dosage sensitivities assessed from large-scale deletion or overexpression screens (Supplementary Fig. 4b).

Similarly, a twofold increase in noise levels results in fitness losses from 0% to 3%, with an average fitness loss of 0.9% (Fig. 4c); and half of all genes (16 out of 33) are significantly noise intolerant (at FDR < 10%, estimated using randomized controls). These results are therefore rare evidence based on experimental data that high noise in the expression of many genes, i.e., short-lived expression fluctuations away from optimal wild-type expression, does impact organismal fitness in yeast.

In line with previous reasoning[12,13,32–34], expression sensitivity and noise intolerance are correlated across genes (Pearson correlation $R = 0.65$, Fig. 4d). This correlation does not arise from an inherent coupling between mean and noise or how we have reconstructed fitness landscapes (permutation test, $p = 0.003$). Importantly, across genes noise intolerance is nearly as large as expression sensitivity, revealing that reducing expression noise and optimizing mean expression should be of similar importance in order to maximize organismal fitness.

Similar conclusions, in terms of effect sizes of expression sensitivity and noise intolerance as well as the significance of effects, are reached if both measures are instead estimated from partial correlations on the raw data of gene-promoter strains (Supplementary Fig. 4c).

Together with previous analyses[12,13,26,32–34], these results suggest that too much noise in the expression of many yeast genes impairs organismal fitness. During evolution, therefore, selection may have acted to minimize noise in the expression of

these noise intolerant genes. To test this, we compared how the noise intolerance quantified on each genes' fitness landscape relates to its measured in vivo protein expression noise in multiple published datasets (Supplementary Fig. 4d). Noise intolerance is indeed negatively correlated with the endogenous protein expression noise of genes in three different large-scale datasets (Spearman rank correlation: $\rho = -0.26$ (noise in YPD), $\rho = -0.29$ (noise in SD), $\rho = -0.43$ (noise diploids), aggregated $p$-values from permutation test using Fisher's method, $p = 0.048$)[12,41]; and this effect is consistent across different metrics of noise intolerance (Supplementary Fig. 4d). Similarly, as expected from a high correlation with noise intolerance, expression-sensitivity is also negatively correlated with endogenous protein expression noise ($\rho = -0.17$, $\rho = -0.21$, $\rho = -0.6$; aggregated $p$-values from permutation test using Fisher's method, $p = 0.053$), though results are less consistent across metrics.

Together, this provides good evidence that selection has acted during the evolution of budding yeast to minimize fluctuations in gene expression due to their detrimental impact on organismal fitness.

**Two principal topologies of expression-fitness landscapes.** We next investigated the reasons why, despite a variety of topologies observed across expression-fitness landscapes (see Fig. 3) and the various molecular functions that the investigated genes are involved in, expression-sensitivity and noise intolerance on fitness landscapes are well correlated. We thus asked whether there are any commonalities between the fitness landscapes by performing a principal component analysis across all landscapes using the 8-fold mean expression range around the predicted wild-type expression of each gene (Supplementary Fig. 5a).

Strikingly, the principal component analysis revealed two dominant topologies, that together explain 96% of the variance across landscapes (Fig. 5a and Supplementary Fig. 5b).

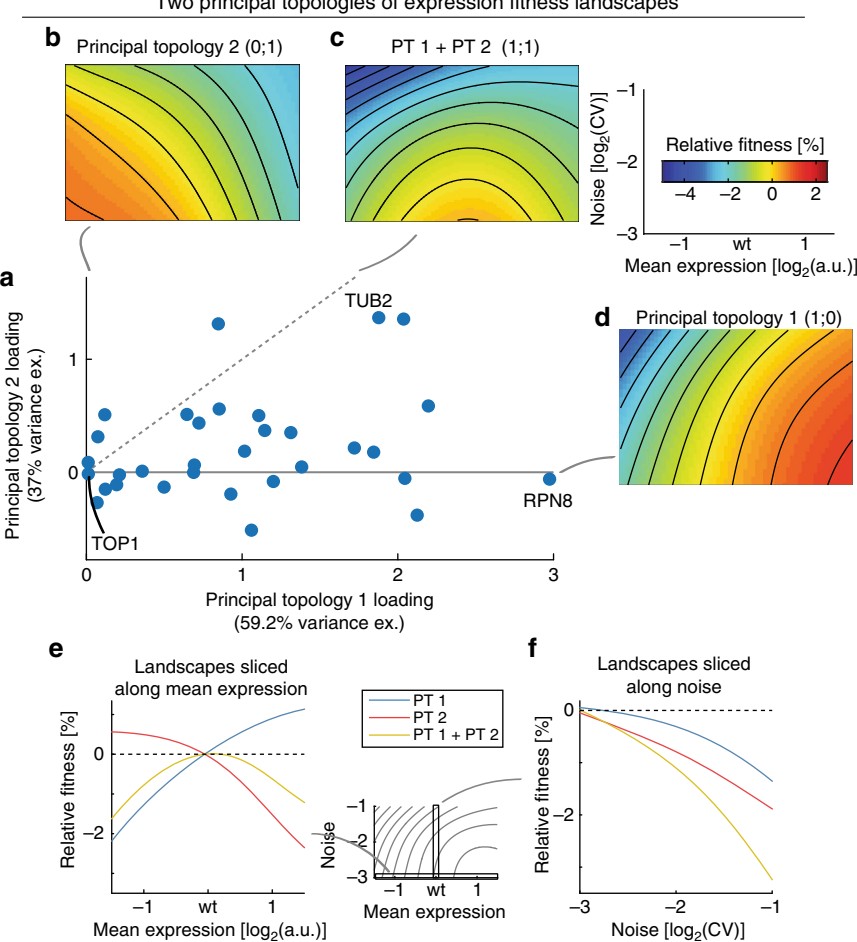

**Fig. 5** Two principal topologies of expression-fitness landscapes. **a** Principal component analysis of expression-fitness landscapes reveals two principal topologies (PT) that explain the majority of variation observed across all landscapes. Scatter plot shows PT loadings of individual landscapes (dashed lines show equivalent loadings with both PTs). Note that results for *RAP1* are not shown and are discussed in the Supplementary Note 1. Source data are provided as a Source Data file. **b–d** Landscapes show individual principal landscape topologies (PT1 and PT2) and their equivalent combination (PT1+PT2). These plots show relative fitness, that is fitness normalized to fitness at wild-type mean expression and low noise ($\log_2(CV) = -3$). Note that relative fitness on principal topologies is an arbitrary scale on its own and only becomes meaningful once multiplied by the respective topology loadings of genes. **e** Principal topologies sliced along mean expression range at minimal expression noise ($\log_2(CV) = -3$). **f** Principal topologies sliced along noise range at wild-type mean expression

Common to both principal topologies is their intolerance for high expression noise (Fig. 5f). Moreover, both topologies show a monotonically saturating relationship between fitness and protein abundance, though with opposing directionality of this relationship (Fig. 5e).

The first principal topology exhibits high fitness if mean expression is at or above wild-type mean expression and if expression noise is low (Fig. 5d). Fitness drops, however, for both lower than wild-type mean expression and high noise. The first principal topology therefore correlates with the fitness consequences of protein shortage.

In contrast, the second principal topology has high fitness at or below wild-type mean expression and at low expression noise, but lower fitness at high mean expression or high noise (Fig. 5b); it therefore correlates with the fitness consequences of protein surplus.

Individual landscapes are made up of different combinations of the two principal topologies (Fig. 5a). All landscapes have positive loadings for the first principal topology, suggesting that the fitness effects of protein shortage are at best neutral but are detrimental for most genes. Indeed, loadings for the first principal topology are predictive of a gene's essentiality (Supplementary Fig. 5c).

Genes show both positive as well as slightly negative loadings for the second principal topology (with one exception, see Supplementary Note 1). Combinations of positive loadings for both topologies lead to peaked landscapes, with decreased fitness and amplified negative impact of high noise when mean expression deviates from wild-type expression in either direction (Fig. 5c). Three genes (*ABF1*, *TUB2* and *PRE2*) show pronounced peaked patterns, consistent with findings of essentiality as well as sensitivity to copy number amplifications for all three genes[17,42,43]. An additional nine genes (*MLC1*, *RPB10*, *RPT2*, *SEC27*, *SEC53*, *SEC61*, *SEC63*, *SPT15* and *TUB1*) show somewhat weaker peaked patterns. While eight of these genes are essential for growth, none of these genes has previously been found to be sensitive to overexpression, suggesting that the patterns observed here might be subtler than those that can be detected by large-scale overexpression screens[42,43].

In summary, two principal topologies in mean-noise expression space—representing the elemental response to having too few or too many proteins—explain nearly all variability in the reconstructed fitness landscapes. Because an individual topology captures a fixed relationship between how changes in mean and

noise affect fitness, the fact that all fitness landscapes are essentially explained by just two topologies explains the observed correlation between fitness effects of short-term (noise) and sustained (mean) deviations from optimal protein abundance across genes.

**Peaked landscapes uncouple the evolution of noise from mean**. Finally, we used the expression-fitness landscapes to explore how gene expression might evolve under the phenotypic constraints imposed on changes in noise and mean expression by the transcriptional process.

In gene expression, mutations in cis-regulatory elements, e.g., the promoter region, have specific effects on mean expression and expression noise that are determined by how they affect the underlying molecular mechanisms of transcriptional bursting[7–11] (Fig. 6a). The molecular mechanisms underlying the transcriptional process thus couple noise and mean expression and constrains how genetic variability can affect both expression phenotypes.

To explore whether the transcriptional process constrains evolutionary trajectories in mean-noise space we simulated adaptive walks on the principal topology landscapes (and their combination). For simplicity, we abstracted adaptive walks such that only steps consistent with the primary cis-regulatory changes found in promoter regions are allowed (Fig. 6a), steps have unit size, their likelihood depends on the potential fitness gain and each grid-point on a fitness landscape represents an accessible genotype (see "Methods"). Moreover, initially we assumed that mutations affecting burst frequency are as likely to occur as mutations affecting burst size (see "Discussion" for outcomes of alternative scenarios).

On both principal topologies, we find that the coupling of noise and mean by the transcriptional process restricts the evolution of noise levels. On principal topology 1 (sensitivity to protein shortage) genes evolve towards higher mean expression levels and lower noise levels. The final noise minimum, however, strongly depends on the noise level of the starting point, as noise cannot be reduced further than what is maximally achieved by always selecting for frequency increasing over size increasing mutations (Fig. 6b). On principal topology 2 (sensitivity to protein surplus) genes evolve towards lower mean expression. Expression noise, however, at best stays constant (if size altering mutations are selected for) or increases (if frequency altering mutations are selected for), thus moving away from optimally low gene expression noise (Fig. 6c). This suggests that, when genes evolve on monotonic, saturating fitness landscapes, the cis-regulatory evolution of gene expression noise is limited by its coupling to mean expression changes.

In contrast to the monotonic principal topologies, evolutionary trajectories on peaked landscapes (PT1+PT2) exhibit a bi-phasic behaviour (Fig. 6d). These trajectories are characterized by a first phase of evolution towards optimal mean expression (potentially with coupled changes in expression noise) and a second phase of evolution towards lower expression noise, during which mean expression levels hardly change. Strikingly, independent of the starting point of the simulations, this second phase occurs in a well-defined, narrow region of the landscape (Fig. 6d).

We find that this region, which we term the noise funnel, is created by a misalignment of the regions where burst frequency and burst size altering mutations are beneficial or detrimental (determined by the points at which equi-fitness lines are tangential to the mutational vectors, Fig. 6e and Supplementary Fig. 6a, b). Specifically, here, mutations that *increase* burst frequency and mutations that *decrease* burst size are beneficial, the combination of which results in lowered expression noise but unaltered mean expression (Fig. 6f). Consistently, evolution towards lower expression noise in the noise funnel proceeds via alternating steps of increased burst frequency and decreased burst size mutations.

Moreover, evolution towards lower expression noise is accelerated by the epistatic interactions—the non-independence of fitness outcomes—between the two opposing mutations. In particular, a mutation of one type renders a consecutive mutation of the same type less beneficial (Fig. 6f), i.e., consecutive mutations of the same type are negatively epistatic due to the saturating relationship between fitness and both mean expression and noise (see Fig. 4f, g). The first mutation does, however, render the alternative mutation more beneficial, i.e., their combination is positively epistatic (Fig. 6f). The noise funnel therefore not only uncouples the evolution of noise from mean expression but accelerates the independent minimization of expression noise levels via the genetic interactions of burst size and frequency modulating mutations.

## Discussion

We reconstructed empirical expression-fitness landscapes that allowed us to systematically investigate the quantitative effects of two molecular phenotypes, mean expression and noise, on organismal fitness in yeast.

Across 33 reconstructed landscapes nearly all variance in fitness profiles is described by linear combinations of only two principal topologies, which represent the fitness effects of having too few or too many proteins. These two principal topologies imply that there exist fundamental functional relationships between protein shortage or surplus and organismal fitness that apply to most genes; and that genes only differ in the magnitude of these relationships.

It has been a long-held assumption that genes that are sensitive to sustained depletion or over-expression of their protein abundances are also sensitive to short-lived, stochastic fluctuations in protein abundances[12,13,24,32–34]. Dedicated experimental tests of this hypothesis, however, had so far remained rare[26], because of the difficulty of independently varying mean expression and noise to quantify the effects of perturbing only one of the two.

Our analyses of how fitness varies across continuous mean-noise fitness landscapes overcomes this limitation, allowing the effects of changes in noise or mean to be examined in isolation as well as in context of each other. This confirmed that the more sensitive organismal fitness is to changes in mean abundances of genes the more intolerant it also is to high expression noise in these abundances. Importantly, on most of the expression-fitness landscapes, the fitness cost of high expression noise is of similar magnitude to that of non-optimal mean expression levels.

There are two important caveats to our analyses of fitness landscapes in mean-noise expression space. The first caveat is that we are lacking estimates of the noise level of endogenous promoters as reference points (similar to the estimated mean expression of endogenous promoters) to judge whether the right range of noise levels is explored to quantify the cost of varying noise levels. For genes whose endogenous promoters have lower noise levels than the range covered by the reconstructed fitness landscapes, the cost of increasing noise (by a fixed factor) would likely be lower than estimated, due to the concavity of the relationship between noise and fitness (Fig. 4f).

The second caveat is that the fitness effects of noise when cells are grown in a stable, glucose-rich laboratory condition might differ from more variable natural environments. Specifically, in more variable environments, the variable expression of certain genes to create phenotypic diversity (bet-hedging) can potentially be beneficial[27–31]. Consistently, stress-related genes have been found to have high expression noise[12,13]. The genes for which we reconstructed fitness landscapes are, however, strongly biased to essential genes that carry out cellular core functions (ribosomal subunits, proteasome, cytoskeleton, trafficking and transcription

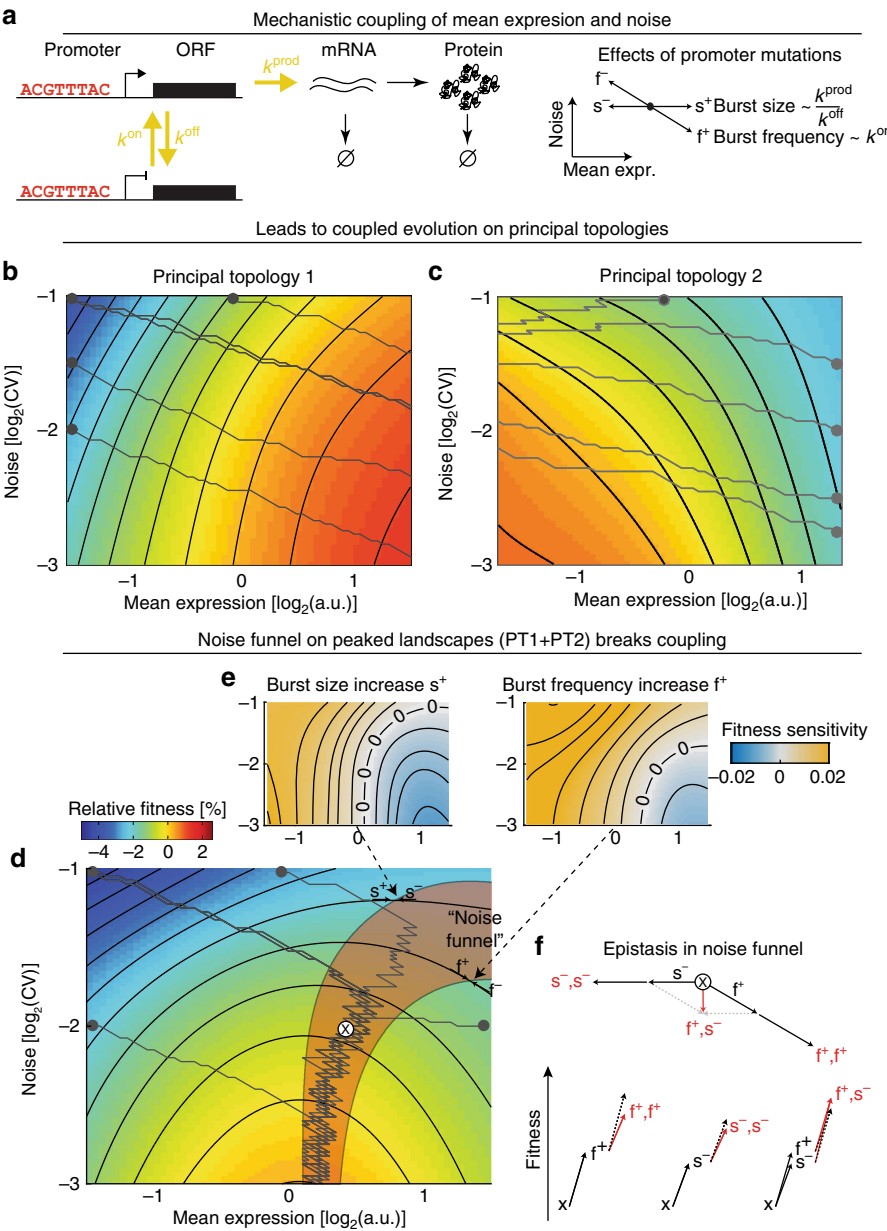

**Fig. 6** Peaked expression-fitness landscapes can break the coupled evolution of mean and noise. **a** Expression of proteins in a two-state telegraph model of transcriptional dynamics. The promoter switches between transcriptionally permissive and prohibitive states, with rates $k^{on}$ and $k^{off}$. In the permissive state, mRNAs are transcribed at rate $k^{prod}$, resulting in transcriptional bursts. mRNAs are then translated to proteins; and both are degraded eventually. Promoter mutations affect transcriptional bursts by changing their size (altering $k^{prod}$ or $k^{off}$) or their frequency (altering $k^{on}$). Mutations that affect burst size result in concordant changes in mean expression, without affecting noise. Mutations that affect burst frequency result in concordant changes in mean expression and opposing changes in noise. **b**–**d** Evolution of gene expression properties via promoter mutations on principal topology 1 (sensitivity to protein shortage, panel **b**), principal topology 2 (sensitivity to protein surplus, panel **c**) or peaked (sensitivity to protein shortage and surplus, panel **d**) fitness landscapes. Each grey trajectory is one realization of a stochastic walk, where the likelihood of steps depends on their fitness gains (circles indicate starting genotype). All grid points on the landscape are assumed to be accessible genotypes. In panel **d**, red patch indicates noise funnel, where burst size-decreasing mutations ($s^-$) and burst frequency-increasing mutations ($f^+$) are beneficial, which results in alternating steps towards lower noise levels. x in noise funnel marks point used to exemplify epistasis in panel **f**. **e** Fitness gains of mutations increasing burst size ($s^+$, left) or increasing burst frequency ($f^+$, right) on peaked landscape. Borders of no gain (contour marked by 0's) do not align due to additional fitness gains from reduced noise in mutations increasing burst frequency, creating the *noise funnel* on the peaked fitness landscape. **f** Upper: Resulting mean/noise effects of different combinations of increased burst frequency ($f^+$) and decreased burst size ($s^-$) mutations. Lower: Fitness and epistasis from mutation combinations. First step: fitness of individual mutations. Second step: Expected fitness (black, dashed) and observed fitness (red, solid) of combined mutations. Epistasis is the difference between observed and expected fitness. Starting genotype x is marked on peaked fitness landscape in panel **d**

factors). Such genes are biased towards low expression noise[12,13,32–34] suggesting that, even in natural (variable) environments, they have to be precisely expressed.

Moreover, our analysis of expression-fitness landscapes was focused on an eight-fold range around wild-type expression levels, which allowed us to reveal systematic fitness effects across many genes. The fitness effects of expression noise are, however, expected to depend on the discrepancy between the actual and optimal average expression levels[44]. In particular, high expression noise should become beneficial when average expression is far away from optimum, as this would allow some cells to transiently express more beneficial protein abundances, therefore increasing overall population growth rate. This has recently been demonstrated for the *TDH3* gene in yeast[26]. Indeed, when examining expression-fitness landscapes initially excluded from our analyses due to wild-type expression levels outside of the investigated mean expression range, we find examples of this transition in noise-fitness effects. For the two highly expressed genes *ENO2* and *RPL3*, high expression noise turns from being detrimental when mean expression is close to wild-type levels to beneficial when mean expression drops far below wild-type expression levels (Supplementary Fig. 7). The fitness at low mean expression and high noise is, however, lower than fitness at more optimal mean expression and low noise. This shows that, while high noise can improve fitness if expression is far from its optimum, it is by no means a substitute for optimally tuned expression levels[45].

We have further used the concept of expression-fitness landscapes to study evolutionary scenarios for gene expression under the phenotypic constraints imposed by the transcriptional process. This revealed that expression noise levels cannot be effectively optimized via *cis*-regulatory evolution for genes that only have sensitivities to either protein shortage or surplus, thus raising the question whether genes with monotonic fitness landscapes have non-optimal noise levels or if and how optimization is achieved in *trans*. In contrast, combined sensitivities to protein shortage and surplus, which one third of the assayed genes display, create a narrow landscape region —the noise funnel—in which the evolution of noise is uncoupled from mean expression. The noise funnel is the consequence of a disagreement in the signs of fitness effects of burst size and burst frequency modulating mutations. The independent evolution of low noise levels is further promoted by genetic interactions between the types of mutations, where the combination of both mutation types are positively epistatic but two consecutive mutations of the same type are negatively epistatic.

We performed these evolutionary simulations under the simplifying assumption that burst size and burst frequency mutations are equally likely to occur. Typically, mutations that change burst frequency are, however, much more likely to occur in promoter regions than mutations affecting burst size[7]. Consistent with the epistatic interplay of both mutation types in the noise funnel, we find that an equal likelihood for both types of mutations to occur is key to rapid reduction of expression noise within the noise funnel (Supplementary Fig. 6c). Evolution of minimal gene expression noise would therefore be hampered if burst size could only change via mutations in the promoter. Changes in post-transcriptional processes, however, also affect the size of expression bursts[11], thus enlarging the mutational target space for burst size changing mutations and potentially accelerating the evolutionary minimization of expression noise level. The vast expansion of post-transcriptional repressive regulators in higher eukaryotes, such as microRNAs, could have therefore facilitated the reduction of gene expression noise across distinct cellular states[46–48]. Consistently, human dosage-sensitive genes are highly enriched for microRNA binding sites[49,50].

Together, this shows that in order to understand the evolution of gene expression both the constraints imposed by the underlying molecular mechanisms as well as the mapping between expression distributions and organismal fitness have to be considered. Moreover, our analysis makes the testable prediction that for peaked genes, regulatory elements with opposing influences on burst size and burst frequencies should co-evolve in order to minimize expression noise.

## Methods

**Fitness calculations.** Relative fitness for growth in glucose of each promoter-gene pair strain was calculated from changes in read count frequencies across the competitive growth experiment[24]. Fitness at two time-points (23 and 35 h growth) were calculated as

$$f_i^t = \log_2\left(\frac{n_i^t/n_{wt}^t}{n_i^{t0}/n_{wt}^{t0}}\right) \qquad (1)$$

with $n$ as the number of reads (supplemented with a pseudo count of 0.1), subscripts denoting strain i or the wildtype wt, and superscripts denoting the time-point ($t0$ as starting time-point of the competition experiment, $t$ for the two later time-points). A linear model was fit to derive a normalization factor to correct systematic fitness differences across all promoter-gene pair strains between the two time-points (Matlab function fit with option poly1, i.e., a first order polynomial with slope and intercept was fit). Fitness for each promoter-gene pair was then calculated as the weighted average of relative fitness measures at both time-points, with weights as the inverse of error estimates calculated from read counts as

$$\sigma_i^t = \sqrt{\frac{1}{n_i^{t0}} + \frac{1}{n_i^t} + \frac{1}{n_{wt}^{t0}} + \frac{1}{n_{wt}^t}} \qquad (2)$$

A combined error of fitness for each promoter-gene pair was accordingly derived as

$$\sigma_i = \left(\left(\sigma_i^{23h}\right)^{-2} + \left(\sigma_i^{35h}\right)^{-2}\right)^{-1/2} \qquad (3)$$

**Promoter expression properties.** Promoter mean expression and promoter expression noise was calculated as average over two replicates[38]. Error of both measures was estimated as the running average of replicate standard deviation as a function of sequencing read based error estimate over all promoters (calculated using MATLAB function fit, with method loess and span 0.5).

**Data pre-processing/quality control.** Promoters were checked for consistency of mean expression estimates between driving *YFP* on a plasmid[38] and driving *YFP* from the *HIS3* locus[24]. A linear model fit to the log₂-transformed mean expression data was used to transform the plasmid-derived data in order to make the two studies comparable (Matlab function polyfit with degree 1, i.e., slope and intercept were fit). Eleven of 120 promoters that showed a log₂-derivation of more than 0.5 between mean expression estimates in both studies were discarded (Supplementary Fig. 1a). Another six promoters that had a median fitness error estimate over all promoter-gene combinations >0.1 were discarded (Supplementary Fig. 1b). Finally, to restrict our analysis to a sufficiently homogenously populated core region in the mean-noise space, 24 promoters with mean expression below 2 or above 6 log₂-expression units were discarded (Supplementary Fig. 1c). Because our subsequent analyses are focused on the fitness effects around the wild-type expression of genes, only those 33 of 85 genes that have an estimated mean expression output of their wild-type promoters that lies in the centre of the analyzed expression range (between 3 and 5 log₂-expression units) were considered (Supplementary Fig. 1d). Additionally, for the transcription factors *ABF1*, *MIG1* and *RAP1* several promoter-gene pairs (3, 5 and 2, respectively) were discarded from our analysis because the promoters contain predicted binding motifs for these genes.

**Calculation of mean-noise fitness landscapes.** To reconstruct a smooth, continuous fitness landscape for each gene, we calculated fitness values on a regular grid across the mean-noise space using a Gaussian smoothing approach. The grid dimensions were chosen such that the rectangular grid covers all promoter strains in the mean noise space and grid points were spaced by 0.05 log₂-mean expression units and 0.025 log₂-noise (CV) units. For subsequent analyses of expression-fitness landscape features, we investigated grids that extend ±1.5 log₂-mean expression units from the wild-type promoter expression and range between −3 and −1 log₂-noise (see below) and thus have $61 \times 81 = 4941$ grid points. For visualization purposes (Fig. 2f) we also computed more extensive grids. For each grid point $xy$, a fitness value was calculated as the weighted average over the fitness of all gene-specific strains. How the strain in which promoter $i$ drives the gene $j$ contributes to fitness at grid point $xy$ was calculated by integrating over the joint probability density function of a Gaussian smoothing kernel centred on the grid point and a Gaussian likelihood function centred on the promoter position in mean-noise space. The Gaussian smoothing kernel is a bi-variate normal density (Matlab function *mvnpdf*) with means $\mu = x$ and $\eta = y$, the grid point position in mean-noise space, and covariance matrix $C = \begin{bmatrix} 0.602^2 & 0 \\ 0 & 0.361^2 \end{bmatrix}$, the optimal

shape of the kernel that minimizes the RMSE between fitness surfaces and measured fitness of promoter-gene strains, as estimated from ten-fold cross validation. The Gaussian likelihood function of the true position of the promoter $i$ in mean-noise space is a bi-variate normal density with means $\mu = \mu_i$ and $\eta = \eta_i$, the estimates of mean expression and noise of the promoter, and covariance matrix $C = \begin{bmatrix} \sigma_{\mu_i}^{-2} & 0 \\ 0 & \sigma_{\eta_i}^{-2} \end{bmatrix}$, the error estimates for mean expression and noise of the promoter. The integral over the joint probability densities, further normalized by the uncertainty of the fitness estimate of promoter-gene strain $ij$, results in the weighting of the fitness of promoter-gene strain $f_{ij}$ for the fitness at grid point $xy$ in the fitness landscape of gene $j$:

$$w_{ij \to xy|j} = \frac{1}{\sigma_{f_{ij}}^2} * \iint PDF_{\text{grid } xy} * PDF_{\text{promoter } i}\,\mathrm{d}\mu\,\mathrm{d}\eta \quad (4)$$

In practice, to speed up computations at little cost to precision, $w_{ij \to xy|j}$ was calculated on a $21 \times 21$ auxiliary grid around the grid point $xy$, with spacing $0.3 * \sqrt{0.602}$ in the mean and $0.3 * \sqrt{0.361}$ in the noise expression direction and only using those auxiliary grid points where the smoothing kernel probability density is larger than 1% of the respective density on the grid point $xy$.

The fitness at grid point $xy$ in the fitness landscape of gene $j$, $f_{xy|j}$, is the weighted average over the fitness values of all gene-specific strains, i.e.

$$f_{xy|j} = \sum_j \left( f_{ij} * w_{ij \to xy|j} \right) \Big/ \sum_j w_{ij \to xy|j} \quad (5)$$

Note that some landscapes have non-optimal fitness at wild-type expression, especially steeper landscapes with asymmetric shapes (Fig. 3), such as *RPN8* (Fig. 2f). These effects might result from two plausible causes that are both due to blurring of fitness landscapes: first, the noise levels assayed by the synthetic promoters might be higher than the noise levels of the endogenous promoters, i.e., at even lower noise levels the actual expression-fitness relationships are sharper and fitness at wild-type expression is optimal, or second, the resolution limit of our smoothing procedure, which in turn is dictated by the experimental errors in the data. Because our study is not concerned with the fine-grained details of individual expression-fitness landscapes but rather with the general patterns across different landscapes, we conclude that these effects should not impact the generality of our results.

**Principal component analysis of fitness landscapes.** To understand whether the expression-fitness landscapes share common topological features we performed a principal component analysis (PCA) across all landscapes (Supplementary Fig. 5). For this analysis, landscapes extending $\pm 1.5$ $\log_2$-mean expression units from each gene's wild-type promoter expression and ranging from $-3$ to $-1$ $\log_2$-noise units were compared between the 33 genes. Prior to performing the PCA, fitness values on each landscape were normalized to the fitness at wild-type expression and $\log_2(\text{noise}) = -3$. PCA was performed with Matlab function pca (option centred set to true) using the fitness on each of the $61 \times 81 = 4941$ grid points of each gene's landscape as observations and treating the 33 genes as variables. Reported principal component loadings (principal topology loadings) for each landscape were corrected for the loadings of the mean fitness landscape (its loadings for the first and second components are 1.03 and 0.15, respectively).

**Calculation and comparison of expression sensitivity and noise intolerance metrics.** Expression sensitivity of each gene was calculated as the average absolute slope of the mean expression-fitness function at $\log_2(\text{noise}) = -3$. It therefore indicates the loss of fitness due to changes in mean expression, no matter the direction. Noise intolerance of each gene was calculated as the average negative slope (first derivative) of the noise-fitness function at wild-type mean expression. It therefore indicates the loss of fitness due to increases of expression noise. For better intuition, we normalized expression sensitivity and noise intolerance to correspond to the fitness loss upon a two-fold change of mean or a twofold increase of noise, respectively. Expression sensitivity and noise intolerance were also computed for $10^4$ randomizations of each gene's fitness landscape, where in each randomization the fitness values between all promoter-gene strains were permutated. $p$-values for each gene's expression-sensitivity and noise intolerance were calculated as the fraction of the gene's randomized fitness landscapes with expression-sensitivity and noise intolerance values greater or equal to the non-randomized values. Positive FDR was calculated from these $p$-values using the linear step-up procedure (Matlab function mafdr with option BHFDR). Additionally, the Pearson correlation between expression sensitivity and noise intolerance across all genes was calculated for each randomization run. A $p$-value for the Pearson correlation coefficient between expression sensitivity and noise intolerance on real landscapes was derived as the fraction of correlation coefficients from randomization runs that are greater or equal than that of the real data.

Expression-sensitivity and noise intolerance were also derived from raw gene-promoter strain data. Here, for each gene the Pearson partial correlation coefficients between noise or mean expression levels and fitness of gene-promoter strains were calculated while controlling for the other expression phenotype (using Matlab function partialcorr). $p$-values for alternative hypothesis that partial correlation is not 0 were used to calculate positive FDR using the linear step-up procedure (Matlab function mafdr with option BHFDR). As for the rest of our analysis we only considered promoters within the expression range of 2–6 $\log_2$-mean expression

units (Supplementary Fig. 1c). Not unexpectedly, correlation between expression-sensitivity and noise intolerance are somewhat smaller, which might stem from the fact that partial correlations can only identify linear dependencies, but e.g. not the peaked expression-fitness relationships of many landscapes.

Moreover, we used two additional expression-sensitivity measures from published data. First, the expression curvature metric used by Keren et al.[24] was calculated as described therein, i.e. as the minimal mean expression distance at which a 5% fitness drop compared to fitness at wild-type expression is observed (on impulse fitted fitness data as reported in Supplementary Table S3 of ref. [24]). Second, we classified genes in a binary fashion as dosage sensitive, if they have been reported to be essential[17] ($n = 23$; $n = 3$ of which are also haplo-insufficient[16]) or over-expression sensitive[42,43] ($n = 11$, nine of which are also essential) in large-scale genetic screens, or dosage-insensitive, if they have not been reported as either essential or over-expression sensitive before.

**Noise tolerance metric comparison to endogenous noise levels.** Noise intolerance and the three metrics of expression sensitivity of genes were compared to endogenous noise levels reported in large-scale screens by calculating the Spearman rank correlation coefficient. $p$-values were derived for the alternative hypothesis that correlation is smaller than 0 and aggregated for the three tests of each metric using Fisher's method. Endogenous noise levels in haploid cells when grown in minimal medium (SD) or rich medium (YPD) for 18 and 22 genes out of the 33 genes investigated here were obtained from Newman et al.[12] and reported noise DM values (deviation from running median) were used for comparison. Additionally, endogenous noise levels in diploid cells for 9 out of the 33 genes were obtained from Stewart-Ornstein et al.[41]. Noise levels in diploid cells were corrected for the running median of noise levels across expression levels, similar to the DM procedure[12].

**Evolution on fitness landscapes.** Evolutionary simulations on fitness landscapes in a promoter mutation scenario were implemented as stochastic walks using a Gillespie algorithm[51]. Here, the probability of a mutation to be selected for is proportional to its fitness gain relative to the summed fitness gains from all mutations with non-negative fitness gains. The time until the next mutation is selected for is exponentially distributed with mean proportional to the inverse sum of non-negative fitness gains. Each step results in the jump to an adjacent grid point. For burst size mutations, this jump is to an adjacent grid point with altered mean expression (plus or minus 0.05 $\log_2$-mean expression units, if an increase or a decreased burst size is selected for, respectively) but equal noise. For burst frequency mutations, this jump is to an adjacent grid point with both altered mean expression (plus or minus 0.05 $\log_2$-mean expression units, if an increase or a decreased burst frequency is selected for, respectively) and altered noise (minus or plus 0.025 $\log_2$-mean expression units, respectively; grid is spaced twice as narrow in noise direction, thus the change in noise for a burst frequency mutation is the negative square root change in mean expression).

To simulate differential likelihoods of mutations (related to Supplementary Fig. 6c), we modified the Gillespie algorithm by altering the calculation of probabilities for mutational selection and time intervals. For example, for the scenario where burst size mutations are ten times less likely, their fitness gains were divided by a factor of ten in the calculation of probabilities, i.e., they were ten times less likely to be selected for.

**Reporting summary.** Further information on research design is available in the Nature Research Reporting Summary linked to this article.

## Data availability
No primary data have been generated in this study. All data sources are listed in Supplementary Table 1. The source data underlying Figs. 2d, 4b–d and 5a are provided as a Source Data file. Pre-processed data can also be found at https://github.com/lehner-lab/mean-noise-fitness-landscapes.

## Code availability
All analysis was performed using Matlab version R2014b. All code to repeat the analyses can be found at https://github.com/lehner-lab/mean-noise-fitness-landscapes.

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

## Acknowledgements

This work was supported by a European Research Council Consolidator grant (616434), the Spanish Ministry of Economy and Competitiveness (BFU2011–26206 and SEV-2012-0208), the AXA Research Fund, Agència de Gestió d'Ajuts Universitaris i de Recerca (AGAUR, 2014SGR831), FP7 project 4DCellFate (277899), the EMBL-CRG Systems Biology Program (all to B.L.), an EMBO Long-Term Fellowship (ALTF 857–2016), the European Union's Horizon 2020 research and innovation programme (Marie Skłodowska-Curie grant agreement No 752809) (both to J.M.S.) an AGAUR grant (2014SGR0974) and a MINECO grant (BFU2015–68351-P) (both to L.B.C.). The authors acknowledge support from the Spanish Ministry of Economy, Industry and Competitiveness (MEIC) to the EMBL partnership, the Centro de Excelencia Severo Ochoa, and the CERCA Programme / Generalitat de Catalunya.

## Author contributions

J.M.S. performed all analyses. J.M.S., L.B.C. and B.L. conceived the study. J.M.S. and B.L. designed analyses and wrote the manuscript with input from L.B.C.

## Additional information

**Competing interests:** The authors declare no competing interests.

