## [Peer Review File · Nature Communications]

Reviewers' comments:

Reviewer #1 (Remarks to the Author):

In the manuscript entitled "Empirical mean-noise fitness landscapes reveal the fitness impact of gene expression noise", Schmiedel and co-authors investigate the effect of gene expression noise of 33 genes on the fitness of budding yeast. To achieve this, the authors construct the fitness landscapes of these genes in mean-noise expression space which allows the uncoupling of the effects of mean expression and expression noise on fitness. Doing so, the authors find that expression noise can be as detrimental to organismal fitness as deviations from optimal mean expression. Finally, they perform evolutionary simulations within mean-noise space to explore the concomitant optimization of both expression mean and noise. Although, it is known that expression noise can have non-trivial effects on fitness and is consequently likely under selection, the quantitative approach used in this study provides a deeper understanding of the relationship between expression noise, expression mean and fitness in an evolutionary framework. As such, this study represents an important step forward and provides clear hypotheses for future experimental research.

Overall, the manuscript is well-written and clearly structured and presented. The methods are sound and the results convincing. The study can make an important contribution and therefore may be suitable for Nature communications. However, I do have a few comments, which I would like to see addressed first.

In the first section of the results, the authors go to great lengths to discuss quality control and their rationale behind obtaining their final dataset. Although the detailed description is much appreciated, I still felt compelled to read the studies that originally obtained this data, which I eventually did. To facilitate the reader, the authors could perhaps concisely describe the specific experiments from which their data was obtained in the supplemental methods.

The authors indicate that about half of the genes in the panel exhibit significant noise intolerance and/or expression sensitivity. However, I am more surprised that half of the genes are noise tolerant and/or expression insensitive. Is this an artefact of the data being not sensitive enough to significantly detect subtle intolerance or sensitivity? Or do these genes represent a specific class which are less sensitive? If so, for these genes minimization of noise (even optimization of expression) would not be beneficial for organismal fitness. This should be addressed by the authors?

Regarding the evolutionary simulations, I find it difficult to think about noise minimization in the context of an adaptive walk (i.e. through positive selection). The phenotypic result of expression noise in the context of a population would be having a phenotypically heterogeneous population. As such, negative selection would play a crucial role in reducing the negative impacts of high expression noise rather than selecting for individuals with an intrinsic low expression noise. The authors should address the role of negative and positive (neutral) selection in the evolution of noise minimization.

The authors refer several times to noise as short-lived fluctuations in expression. What do they exactly mean with short-lived? What is the timescale of these short-lived fluctuations and is it different to long-lived stochastic on-off fluctuations?

Reviewer #2 (Remarks to the Author):

In this study, Schmiedel et al. characterized an expression-noise fitness landscape for 33 genes in the budding yeast. This is a big improvement since previous fitness landscapes were largely

characterized as a function of expression level. They analyzed the data quite smart, the results are beautiful, and the presentation is clear. Overall I think it is a nice piece of work. I do have some suggestions that the authors should consider.

Major comments:

1. It is not clear to me why noise was not measured in Fig. 2b. If the flow cytometry data for each promoter are available, the authors should be able to quantify noise here.
2. Rather, the authors measured expression level and noise with another library (Fig. 2c). It is not clear how the copy number variation in plasmid would affect the measurement. It is also unclear how the mean expression levels are combined from two different strategies (fig. 2b and 2c) since the 120 synthetic promoters were measured twice.
3. The Tub2 landscape in fig. 3d is somewhat unexpected. To my knowledge, the overexpression of tub2 to 2-fold is either lethal or very sick. But in fig. 3d it several variants seem to survive very well.
4. The above observation makes me start to worry about the data quality. And I then want to know if the results were replicable.
5. In fig. 2d it is also a bit unexpected that noise and expression levels are not correlated since we know that many mutations affect burst frequency.
6. Figure 6 of the paper looks too speculative. The simulation has several big assumptions. For example, do they consider the effect of each genuine point mutation during the simulation? Some phenotypes may not exist at all in the genotypes space. In other words, the density of genotypes in the landscape should be considered. To conclude noise funnel, an experimental evolution should be performed. I would suggest the authors remove this section as a whole (at least move to the discussion) during revision.

Minor comments:

1. The question mark after the word "fitness" in figure 1b is a bit confusing. Only the question mark after "evolution" should be enough.
2. Page 12 line 266, fig. 4b?
3. Although the authors attempted to explain the observation of RAP1, I am not very much convinced. But I don't know how to explain either. It looks like an outlier (experimental error?) to me. Again, is the result of RAP1 replicable?

Reviewer #3 (Remarks to the Author):

In this study, Schmiedel and colleagues investigate the fitness effect associated with variations in expression level and expression noise. They find that, in their setup, optimal fitness is reached when genes are expressed at their optimal level and with minimal noise. In addition, they also predict that evolution of expression levels follow a more or less defined trajectory, with first optimization of mean expression followed by minimizing noise in a "funnel" towards the optimal state.

Overall, this is an extremely interesting and solid study that expands our understanding of the fitness effects of variations in expression and expression noise. To me, the most interesting facet is the quantification of the effect of both these parameters.

Perhaps the most important restriction of the study is that the researchers use data from cells growing in stable lab conditions in rich medium. It is perhaps not surprising that in these circumstances, optimal fitness is reached when genes are expressed at their optimal level with zero noise (i.e. all cells in the population express each gene at exactly the optimal level, all the time). However, theory predicts that noise may be beneficial in variable and unpredictable environments. While I think that investigating how the findings would hold in more variable (and perhaps more realistic) environments would go well beyond the scope of this study, I would suggest that the authors discuss the restriction of stable conditions more explicitly and in more

detail.

Figure 2. Would it make sense to use bootstrapping to measure and report the accuracy of the predicted, smoothed fitness landscapes (by repeated cycles of leaving out one datapoint, making the landscape and then testing the accuracy of the predicted values for the one datapoint)?

Figure 3. The noise and mean expression of the genes at the top of the figure seem to not show any influence on fitness. Are these genes that are not needed in glucose growth (and thus expressed at very low levels in WT strains), but also not overly detrimental or costly to express?

The authors find that high noise is as detrimental as deviations from optimal expression. Yet, in the fitness landscapes, the color seems to vary more intensely with the horizontal axis (mean expression) and in Figure 3 4B, most datapoints seem to lie below the 45 degree line (and thus show higher sensitivity to expression variability compared to noise)...

Minor comments

Throughout text, italicize gene names (ABC1)

Abstract, lines 27-30 (Sensitivity to both....noise minimisation"). This sentence is unclear without having read the paper and thus perhaps not ideal for an abstract. Consider revising and simplifying.

Intro, line 54: expression distribution across individuals, or across time, or ...?

Intro lines 106-108: this sentence is not clear; consider revising.

Results line 150- "moments" ?

Line 175: remove one ",,"

Line 364: is it possible that for RAP1, the promoter used for strong overexpression is simply not strong enough to go over the optimum? What is the normal expression level of RAP1?

Lines 399 and further. Is it really a "noise funnel" or rather a "mean expression funnel"? The funnel is restricted by mean expression, not by noise, right?

Line 526: I would argue that the more important caveat is the use of stable conditions, see major comment #1.

Please note: Tracked text changes in the manuscript are highlighted in red (insertions or deletions) or blue (moved sections). Below, referee comments have been numbered for structural clarity and our responses are highlighted in green.

Reviewer #1 (Remarks to the Author):

In the manuscript entitled “Empirical mean-noise fitness landscapes reveal the fitness impact of gene expression noise”, Schmiedel and co-authors investigate the effect of gene expression noise of 33 genes on the fitness of budding yeast. To achieve this, the authors construct the fitness landscapes of these genes in mean-noise expression space which allows the uncoupling of the effects of mean expression and expression noise on fitness. Doing so, the authors find that expression noise can be as detrimental to organismal fitness as deviations from optimal mean expression. Finally, they perform evolutionary simulations within mean-noise space to explore the concomitant optimization of both expression mean and noise. Although, it is known that expression noise can have non-trivial effects on fitness and is consequently likely under selection, the quantitative approach used in this study provides a deeper understanding of the relationship between expression noise, expression mean and fitness in an evolutionary framework. As such, this study represents an important step forward and provides clear hypotheses for future experimental research. Overall, the manuscript is well-written and clearly structured and presented. The methods are sound and the results convincing. The study can make an important contribution and therefore may be suitable for Nature communications.

We thank the reviewer for this positive assessment of our work.

However, I do have a few comments, which I would like to see addressed first.

1.1. In the first section of the results, the authors go to great lengths to discuss quality control and their rationale behind obtaining their final dataset. Although the detailed description is much appreciated, I still felt compelled to read the studies that originally obtained this data, which I eventually did. To facilitate the reader, the authors could perhaps concisely describe the specific experiments from which their data was obtained in the supplemental methods.

These experiments are described in some detail in the first paragraphs of the results (now extended):

‘We obtained data on the fitness of yeast strains where in each strain one of a panel of 85 genes is driven by one of a panel of 120 synthetic promoters²³. Here, in one set of experiments, the library of 120 synthetic promoters was cloned upstream of each of 85 open reading frames, replacing the endogenous promoter (Fig. 2a). All constructed strains were pooled and their fitness (growth rate in glucose) was measured in competitive growth experiments. In a second

set of experiments, the synthetic promoters as well as the endogenous promoters of all investigated genes were cloned in front of *YFP* in the *HIS3* locus and flow cytometry was used to determine their relative mean expression strength (Fig. 2b). Together, this allowed the authors to analyze the fitness effects of mean expression changes relative to the wild-type expression of genes²³.

In addition to this dataset, we also obtained data from an earlier study³⁷ from the same group of authors that measured both mean and cell-to-cell variation ('noise', coefficient of variation, i.e. the standard deviation divided by the mean) in the expression of the same set of synthetic promoters driving *YFP* on a plasmid (Fig. 2c). This was achieved by sorting cells along the overall expression distribution, reconstructing individual promoter expression distributions from deep sequencing of sorted cell populations and quantifying their mean and noise.

When combined, these data allow us to not only assess how the mean but also the shape (as quantified here by mean and noise) of protein abundance distributions affects fitness by comparing strains in which different promoters drive the same gene.'

1.2. The authors indicate that about half of the genes in the panel exhibit significant noise intolerance and/or expression sensitivity. However, I am more surprised that half of the genes are noise tolerant and/or expression insensitive. Is this an artefact of the data being not sensitive enough to significantly detect subtle intolerance or sensitivity? Or do these genes represent a specific class which are less sensitive? If so, for these genes minimization of noise (even optimization of expression) would not be beneficial for organismal fitness. This should be addressed by the authors?

The FDR 10% cutoff that we chose to categorize genes as expression sensitive or noise intolerant lies at slightly below 1% fitness effect for a two-fold change/increase in mean expression or noise. However, much smaller fitness defects are likely to be selected against in natural yeast populations (estimated as selection coefficients of $\sim 10^{-7}$ (Wagner, *Mol. Biol. Evol.* 2005). Therefore, even smaller fitness defects due to noise are likely selected against in natural populations.

Moreover, as shown in Supplementary Figure 4b of our initial manuscript, expression-sensitivity as determined from our fitness landscapes is associated with essentiality and over-expression sensitivity measures from previous large-scale studies, such that genes determined to be neither essential nor over-expression sensitive have lower expression-sensitivity (area under the curve (AUC) = 0.76, $p = 0.016$, one-sided Wilcoxon rank sum test).

We have now added the same comparison for noise-intolerance to Supplementary Figure 4b, which shows the same trend: genes previously determined to be neither essential nor over-expression-sensitive are more tolerant to high noise (AUC = 0.72, $p = 0.032$, one-sided Wilcoxon rank sum test).

1.3. Regarding the evolutionary simulations, I find it difficult to think about noise minimization in the context of an adaptive walk (i.e. through positive selection). The phenotypic result of expression noise in the context of a population would be having a phenotypically heterogeneous population. As such, negative selection would play a crucial role in reducing the negative impacts of high expression noise rather than selecting for individuals with an intrinsic low expression noise. The authors should address the role of negative and positive (neutral) selection in the evolution of noise minimization.

The evolutionary simulations concern an adaptive walk from low to high fitness. Therefore, the changes in both mean and noise in expression are due to positive selection acting on mutations that give higher fitness. We acknowledge that there are many assumptions and simplifications in these simulations – as stated in the text – but they are meant to illustrate how these empirical measurements of the fitness effects of changes in mean and noise expression can result in rather unexpected evolutionary dynamics.

1.4. The authors refer several times to noise as short-lived fluctuations in expression. What do they exactly mean with short-lived? What is the timescale of these short-lived fluctuations and is it different to long-lived stochastic on-off fluctuations?

Decay rates for stochastic fluctuations have been found to be on the time-scale of one to two cell cycles, as now noted in the introduction (Sigal2006).

Reviewer #2 (Remarks to the Author):

In this study, Schmiedel et al. characterized an expression-noise fitness landscape for 33 genes in the budding yeast. This is a big improvement since previous fitness landscapes were largely characterized as a function of expression level. They analyzed the data quite smart, the results are beautiful, and the presentation is clear. Overall I think it is a nice piece of work.

We thank the reviewer for this positive assessment of our work.

I do have some suggestions that the authors should consider.

Major comments:

2.1. It is not clear to me why noise was not measured in Fig. 2b. If the flow cytometry data for each promoter are available, the authors should be able to quantify noise here.

The data used in this study have been published by another lab. The focus of the Keren et al., 2016 study was on how mean expression of genes affects organismal fitness and the authors unfortunately did not report noise data in this context.

2.2. Rather, the authors measured expression level and noise with another library (Fig. 2c). It is not clear how the copy number variation in plasmid would affect the measurement. It is also unclear how the mean expression levels are combined from two different strategies (fig. 2b and 2c) since the 120 synthetic promoters were measured twice.

The noise data from Sharon et al., 2014, have been corrected for extrinsic variations (e.g. copy number variation) by computing expression noise from the ratio of intensities of two fluorescent proteins expressed from the same plasmid, one driven by the promoter of interest, the other one driven by a generic promoter to report on cell-specific expression potency (e.g. due to cell state or number of plasmids present per cell).

As shown in Supplementary Figure 1a, relative mean expression levels across the synthetic promoters between the two studies are nearly identical ($R^2 = 0.99$ for the promoters used in our study). Because our study is only concerned with relative mean expression levels on an arbitrary log-scale, measurements from the two studies are interchangeable.

2.3. The Tub2 landscape in fig. 3d is somewhat unexpected. To my knowledge, the overexpression of tub2 to 2-fold is either lethal or very sick. But in fig. 3d it several variants seem to survive very well. The above observation makes me start to worry about the data quality. And I then want to know if the results were replicable.

Fig. 3d shows that, at low noise levels, overexpression of Tub2 does indeed incur a substantial fitness cost, consistent with the literature.

Concerning the data quality, the measurements from individual strains (individual data points) are subject to uncertainty: on average 6% relative error in fitness measurements (including a generic 5% error term that we added during pre-processing in order to account for not reported replicate error), 5% relative error in mean expression direction and 11% relative error in expression noise direction (see Supplementary Figure 1b,e,f). These unavoidable errors in individual measurements are one of the reasons why we computed the fitness landscapes, in order to have reliable estimates of fitness aggregated from a multitude of individual measurements.

Each expression-fitness landscape is the result of a weighted averaging over ~78 data points (with weighting according to the experimental errors of mean, noise and fitness measurements); the resulting fitness landscapes therefore have lower error in the fitness dimension than individual measurements. This is shown in the data presented in Supplementary Figure 3, where we had recalculated each expression-fitness landscape one hundred times by drawing each strain's mean, noise and fitness values from normal distributions according to their experimental estimates and associated errors. We then calculated the standard deviation of fitness values at each grid point on the landscapes (their uncertainty). In response to a request by Referee 3 we have further included a bootstrapping procedure into this resampling (therefore adding more uncertainty to our estimates), where we draw in each resampling 79 promoters with replacement that are then used to calculate the fitness landscape.

The results show that the average uncertainty of fitness values on landscapes is about 1% (measured by the standard deviation of fitness values upon resampling), in accordance with our power to detect expression-sensitivity and noise intolerance at slightly below 1% for two-fold changes of mean or noise (at 10% false discovery rate).

The majority of landscapes, however, shows systematic variation of fitness levels that are greater than this 1% uncertainty, thus showing that our results are reliable and robust to measurement uncertainty in individual strains and that we can detect systematic differences between fitness landscapes using our approach (Supplementary Figure 3).

2.4. In fig. 2d it is also a bit unexpected that noise and expression levels are not correlated since we know that many mutations affect burst frequency.

It is true that single mutations affect expression and noise in a correlated manner, as stated in our manuscript. The synthetic promoters assayed here, however, were deliberately designed to systematically vary in 'affinity, location, spacing and number of several different regulatory elements' (Sharon et al., Nature Biotechnology, 2012, Sharon et al., Genome Research, 2014). In the entire set of promoters, any individual pair will differ from each other by many alterations, enough to break this mechanistic coupling.

2.5. Figure 6 of the paper looks too speculative. The simulation has several big assumptions. For example, do they consider the effect of each genuine point mutation during the simulation? Some phenotypes may not exist at all in the genotypes space. In other words, the density of genotypes in the landscape should be considered. To conclude noise funnel, an experimental evolution should be performed. I would suggest the authors remove this section as a whole (at least move to the discussion) during revision.

We have better explained the caveats of these simulations in the revised manuscript. However, we don't think it is journal policy to present data only in the Discussion section.

The corresponding Result section paragraph describing the nature of the simulations performed now reads:

"To explore whether the transcriptional process restricts evolutionary trajectories in mean-noise space we simulated adaptive walks on the principal topology landscapes (and their combination). For simplicity, we abstracted adaptive walks such that only steps consistent with the primary cis-regulatory changes found in promoter regions are allowed (Fig. 6a), steps have unit size, their likelihood depends on the potential fitness gain and each grid-point on a fitness landscape represents an accessible genotype (see Methods)."

Minor comments:

2.6. The question mark after the word "fitness" in figure 1b is a bit confusing. Only the question mark after "evolution" should be enough.

Changed.

2.7. Page 12 line 266, fig. 4b?

Thank you, changed.

2.8. Although the authors attempted to explain the observation of RAP1, I am not very much convinced. But I don't know how to explain either. It looks like an outlier (experimental error?) to me. Again, is the result of RAP1 replicable?

We agree that the *RAP1* results warrant further experimental investigation and due to the likelihood of this being an experimental error, and similar concerns by Referee 3, we have moved discussion of the *RAP1* results to a Supplementary Note.

Reviewer #3 (Remarks to the Author):

In this study, Schmiedel and colleagues investigate the fitness effect associated with variations in expression level and expression noise. They find that, in their setup, optimal fitness is reached when genes are expressed at their optimal level and with minimal noise. In addition, they also predict that evolution of expression levels follow a more or less defined trajectory, with first optimization of mean expression followed by minimizing noise in a “funnel” towards the optimal state.

Overall, this is an extremely interesting and solid study that expands our understanding of the fitness effects of variations in expression and expression noise. To me, the most interesting facet is the quantification of the effect of both these parameters.

We thank the reviewer for this positive assessment of our work.

3.1. Perhaps the most important restriction of the study is that the researchers use data from cells growing in stable lab conditions in rich medium. It is perhaps not surprising that in these circumstances, optimal fitness is reached when genes are expressed at their optimal level with zero noise (i.e. all cells in the population express each gene at exactly the optimal level, all the time). However, theory predicts that noise may be beneficial in variable and unpredictable environments. While I think that investigating how the findings would hold in more variable (and perhaps more realistic) environments would go well beyond the scope of this study, I would suggest that the authors discuss the restriction of stable conditions more explicitly and in more detail.

We agree with the reviewer that stable lab conditions represent a simplified environment compared to the more variable and unpredictable natural environments that yeast will encounter in the wild. This will transform the static fitness landscapes to dynamic fitness ‘seascapes’ (Mustonen & Lässig, Trends in Genetics, 2009), i.e. fitness landscapes that change across conditions and time for certain genes, e.g. stress-related genes that create phenotypic diversity that can preempt sudden environmental changes ('bet-hedging', as also discussed in the Introduction of our manuscript).

With respect to 'bet-hedging', in our manuscript we discussed how two genes, which are expressed far below their optimal wild-type expression, show increased fitness when they are variably expressed (Supplementary Figure 7). The genes for which we reconstructed fitness landscapes are, however, strongly biased to cellular core components (ribosomal subunits, proteasome, cytoskeleton, trafficking, metabolism, transcription factors). Previous findings that cellular core genes, but not stress-related genes, have low expression noise (Fraser et al., PLoS Biology, 2004; Newman et al., Nature, 2006; Bar-Even et al., Nature Genetics, 2006; Batada&Hurst, Nature Genetics, 2007; Lehner, MSB, 2008), as well as that promoter polymorphisms increasing noise in the expression of the gene *TDH3* (Metzger et al., Nature, 2015) have been

selected against, suggest that, even in natural (variable) environments, core genes have to be precisely expressed for cells to be fit.

We therefore expected that for most genes assayed here, the results derived from stable lab conditions should largely translate to more variable environments.

We have discussed this as a potential caveat in the Discussion section.

3.2. Figure 2. Would it make sense to use bootstrapping to measure and report the accuracy of the predicted, smoothed fitness landscapes (by repeated cycles of leaving out one datapoint, making the landscape and then testing the accuracy of the predicted values for the one datapoint)?

In our initial submission of the manuscript, results for such an analysis were shown in Supplementary Figure 3. Here, we re-sampled fitness landscapes 100 times where in each run mean, noise and fitness of each strain measurement for each gene were drawn from a normal distribution with standard deviation according to the determined experimental error interval.

We have now updated this analysis to also incorporate promoter bootstrapping (i.e. sample with replacement the ~78 promoters to build the landscape, which introduces more variation than leave-one-out sampling). The results show that the average uncertainty of fitness values on landscapes is about 1% (measured by the standard deviation of fitness values upon resampling), in accordance with our power to detect expression-sensitivity and noise intolerance at slightly below 1% for two-fold changes of mean or noise (at 10% false discovery rate).

The majority of landscapes, however, show systematic variation of fitness levels that are greater than this 1% uncertainty, thus showing that we can detect systematic differences between fitness landscapes using our approach (Supplementary Figure 3).

3.3. Figure 3. The noise and mean expression of the genes at the top of the figure seem to not show any influence on fitness. Are these genes that are not needed in glucose growth (and thus expressed at very low levels in WT strains), but also not overly detrimental or costly to express?

Indeed, as shown in Supplementary Figure 4b of our initial manuscript, expression-sensitivity as determined from our fitness landscapes is associated with essentiality and over-expression sensitivity measures from previous large-scale studies, such that genes determined to be neither essential nor over-expression sensitive have lower expression-sensitivity (area under the curve (AUC) = 0.76, $p = 0.016$, one-sided Wilcoxon rank sum test).

We have now added the same comparison for noise-intolerance to Supplementary Figure 4b, which shows the same trend: genes previously determined to be neither essential nor over-expression-sensitive have lower noise intolerance (AUC = 0.72, $p = 0.032$, one-sided Wilcoxon rank sum test).

However, this does not mean these genes are lowly expressed. The wild-type mean expression level of all genes investigated in our study is restricted to a four-fold mean expression window (Supplementary Figure 1d) and we found no

significant relationship between wild-type expression level of genes and their expression-sensitivity (Pearson's $R = -0.12$, $p = 0.5$) or noise intolerance (Pearson's $R = 0.08$, $p = 0.6$).

3.4. The authors find that high noise is as detrimental as deviations from optimal expression. Yet, in the fitness landscapes, the color seems to vary more intensely with the horizontal axis (mean expression) and in Fig 3 4B, most datapoints seem to lie below the 45 degree line (and thus show higher sensitivity to expression variability compared to noise) ...

The fact that color varies more intensely along the horizontal than the vertical axis is in part because the ranges are different: 8-fold on the horizontal 'mean'-axis, only 4-fold on the vertical 'noise'-axis.

That said, sensitivity to mean expression deviations is on average slightly larger than intolerance of high noise. Our general statement is therefore that these two quantities are of 'similar magnitude', and we have qualified this now by stating that noise intolerance is 'nearly as detrimental' as expression sensitivity in the Result section of the manuscript.

Minor comments

3.5. Throughout text, italicize gene names (*ABC1*)

Changed.

3.6. Abstract, lines 27-30 (Sensitivity to both...noise minimisation"). This sentence is unclear without having read the paper and thus perhaps not ideal for an abstract. Consider revising and simplifying.

Thank you, we revised this only stating that certain types of fitness landscapes can break the mechanistic coupling between mean and noise, thus promoting their independent optimization.

3.7. Intro, line 54: expression distribution across individuals, or across time, or ...?

Across time and individuals in an isogenic population, as mentioned in the preceding, first paragraph of the Introduction. We simplified this paragraph to focus on the description of protein abundance distributions and the coupling between mean and noise.

3.8. Intro lines 106-108: this sentence is not clear; consider revising.

Thank you, we simplified this sentence.

"We further use the expression-fitness landscapes to explore how mean and noise can evolve, given their mechanistic coupling imposed by the transcriptional process. We find that on landscapes of genes sensitive to both

protein shortage and surplus the mechanistic coupling between mean and noise is broken, therefore allowing for the independent minimization of noise levels."

3.9. Results line 150- "moments" ?

We have simplified this to "these data allow us to not only assess how the mean but also the shape (as quantified *here by mean and noise*) of protein abundance distributions".

3.10. Line 175: remove one “,”

Thanks.

3.11. Line 364: is it possible that for *RAP1*, the promotor used for strong overexpression is simply not strong enough to go over the optimum? What is the normal expression level of *RAP1*?

The estimated wild-type mean expression level of *RAP1* is the center of the displayed mean expression region (for all genes, the expression-fitness landscapes are centered on the respective wild-type mean expression). Due to similarity of the *RAP1* fitness landscapes with the two fitness landscapes of *ENO2* and *RPL3*, which do indeed have wild-type mean expression higher than the range covered by our fitness landscapes, it is likely that *RAP1* wild-type expression was misestimated or that specific experimental conditions result in disagreement between wild-type expression and optimal expression levels. Given the likelihood that the *RAP1* results are experimental errors and similar concerns by Referee 2, we have decided to move the discussion of *RAP1* to a Supplementary Note.

3.12. Lines 399 and further. Is it really a “noise funnel” or rather a “mean expression funnel”? The funnel is restricted by mean expression, not by noise, right?

The funnel is ‘funneling noise’ i.e. ‘funneling changes in noise’ so we think the term is appropriate.

3.13. Line 526: I would argue that the more important caveat is the use of stable conditions, see major comment #1.

We agree and now discuss this caveat as well.

REVIEWERS' COMMENTS:

Reviewer #1 (Remarks to the Author):

Response to revisions: "Empirical mean-noise fitness landscapes reveal the fitness impact of gene expression noise".

The authors have satisfactorily responded to all my concerns and made the necessary changes throughout the manuscript.

Concerning the evolutionary simulations, although I still feel that this section is rather speculative I do understand the authors perspective of including this into the manuscript. As an experimentalist, simulations will always feel as oversimplifications (which they objectively are), however they do provide important insights as in this case the noise funnel model. So given the authors' changes of this section further acknowledging the simplicity of their simulations and thusly the conclusions drawn therefrom, I feel this section does add value to the manuscript.

Reviewer #2 (Remarks to the Author):

I am satisfied with the revision and response.

Reviewer #3 (Remarks to the Author):

The authors have addressed all my concerns and answered all questions. I would like to congratulate them on a nice piece of work.

REVIEWERS' COMMENTS:

Reviewer #1 (Remarks to the Author):

Response to revisions: "Empirical mean-noise fitness landscapes reveal the fitness impact of gene expression noise".

The authors have satisfactorily responded to all my concerns and made the necessary changes throughout the manuscript.

Concerning the evolutionary simulations, although I still feel that this section is rather speculative I do understand the authors perspective of including this into the manuscript.

As an experimentalist, simulations will always feel as oversimplifications (which they objectively are), however they do provide important insights as in this case the noise funnel model. So given the authors' changes of this section further acknowledging the simplicity of their simulations and thusly the conclusions drawn therefrom, I feel this section does add value to the manuscript.

Reviewer #2 (Remarks to the Author):

I am satisfied with the revision and response.

Reviewer #3 (Remarks to the Author):

The authors have addressed all my concerns and answered all questions. I would like to congratulate them on a nice piece of work.

Authors response:

We thank all three Referees for their effort and the constructive review process.